# Understanding Melanoma Talk on Twitter: The Lessons Learned and Missed Opportunities

**DOI:** 10.3390/ijerph191811284

**Published:** 2022-09-08

**Authors:** Basma T. Gomaa, Eric R. Walsh-Buhi, Russell J. Funk

**Affiliations:** 1School of Public Health, Indiana University, 1025 E 7th St., Bloomington, IN 47405, USA; 2Carlson School of Management, University of Minnesota, 321 19th Avenue South, Minneapolis, MN 55455, USA

**Keywords:** melanoma, twitter, social network analysis, public health, social media

## Abstract

Background: Melanoma is the third most common cause of cancer and the deadliest form of skin cancer among 17–39 year-olds in the United States. Melanoma is a critical public health issue with a substantial economic burden. Cases and associated burdens, however, could be prevented with a greater awareness of, and interventions related to, skin cancer and melanoma-related preventive behaviors. In fact, as social media use is close to ubiquitous, it represents a potential communication modality. However, more research is needed to understand the current state of melanoma-related information exchanged between Twitter users. This study aimed to understand the different types of users controlling the melanoma-related information diffusion and conversation themes on Twitter. Methods: Tweets (*n* = 692) were imported from Twitter between 1 and 31 May 2021 using the Twitter public API; and uploaded to NodeXL to conduct a social network analysis. Results: Health professionals and organizations with medical backgrounds were the main content producers, disseminators, and top influencers. However, information diffusion is slow and uneven among users. Additionally, conversations lacked a focus on preventive behaviors. Conclusion: Twitter is a potential platform for the targeted outreach of individuals in melanoma awareness campaigns. This study provides insights maximizing the effectiveness of Twitter as a communication modality. Our findings can help guide the development of customized content and interventions during melanoma awareness campaigns.

## 1. Introduction

The number of skin cancer cases and melanoma, specifically, is on the rise worldwide [1,2]. Melanoma is ranked as the third most prevalent and deadliest form of cancer among young adults between 15 and 39 years of age in the United States [3]. Indoor tanning among teens and young adults is associated with an increased risk of melanoma; and non-melanoma cases, such as basal and squamous cell carcinomas [4,5]. In addition to presenting as a major public health risk, melanoma also creates a substantial economic burden. The average yearly cost of skin cancer treatment in the U.S. increased from $3.6 billion in 2002–2007 to $8.1 billion in 2007–2011; representing a 126.2% increase (the annual total cost of all other forms of cancer increased by 25.1%) [6].

The potential influence of social media platforms among teens and young adults may be tremendous. Over 90% of individuals this age spend more than 9 h/day on social media platforms, especially on Facebook and Twitter; which are also popular among indoor tanning users [7,8]. However, social media platforms could serve as a potential venue to increase public awareness of skin cancer by targeting individuals of this age [9,10].

Previous studies:

According to Jhawar et al., Twitter is the most reachable platform when compared to other platforms; as illustrated in the “Don’t Fry Day” campaign [9,11]. However, to the best of our knowledge, there are only a handful of studies regarding the use of Twitter in raising skin cancer awareness. Silva et al. reported that people in Australia used Twitter to share their sun-related advice and skin cancer experiences [12]. Another study showed that Twitter is mostly used by individuals to share their tanning bed burn experiences [13]. Additionally, previous research has illustrated an association between the greater odds of indoor tanning and the regular use of Twitter and Instagram [7]. However, all of the above-mentioned studies used content analysis techniques. Social network analysis (or SNA) was employed in a recent study to identify influencers in the skin cancer community on Twitter [14]. In this study, we aim to use the SNA technique to explore the different users’ roles in information flow. The next section describes previous research employing SNA, the methodology chosen for this research, in the health domain.

Background on social network analysis (SNA):

Social media is built on the engagement between users, which appears in different forms (e.g., replying, following, sharing, retweeting, and friending). When users on Twitter interact with each other, for example, they form connections that evolve into complicated network structures. The SNA technique is focused on describing the relations between users of social media; how it impacts the diffusion of information; and the relationship between information diffusion, and the spread of a specific behavior from a societal perspective [15,16].

The implementation of SNA techniques in diverse public health topics has grown, mainly when it comes to describing health behavior or information dissemination patterns [17,18,19]. It is also noteworthy to mention the role of applying SNA to understand the content landscape and the public perspective during COVID-19 from a societal approach [20,21,22].

In sum, and according to the American Cancer Society, three million cases of skin cancer could be avoided annually if individuals were more aware of, and educated regarding, the risks of skin cancer and its associated prevention methods [23]. Social media platforms, especially Twitter, could serve as a potential platform to raise public awareness [24]. However, according to two recent reviews, more research is needed to maximize the effectiveness of social media platforms to prevent skin cancer [10,25]. To the best of our knowledge, the current line of research lacks an understanding of the diffusion pattern of information associated with melanoma specifically in addition to the characteristics of the users controlling its content on Twitter.

We propose the following research questions for the current study: (1) who are the top influencers, information sources, and disseminators for melanoma content on Twitter? (2) what are the main melanoma-related topics discussed on Twitter? and what are the sources of melanoma-related information shared among users?

## 2. Materials and Methods

Data were collected from Twitter using the application programming interface (API) from the Observatory on Social Media (OSoMe) [26]; this allows for locating tweets from the Twitter Decahose, a 10% random sample of all public tweets (not Twitter users). We included tweets in the English language and containing the keywords *melanoma* AND *# melanoma* between 1–31 May 2021 (Melanoma and Skin Cancer Awareness month). A list of tweetid was uploaded to NodeXL [27]. The data set was composed of a total number of *n* = 692 tweets and 724 users. The number of tweets collected may be relatively small compared to other health conditions. However, when compared to a recent surveillance skin cancer study on Twitter [14], the number of tweets and users that we collected are bigger; (tweets *n* = 692 vs. 385) and users’ numbers (*n* = 724 vs. 324). We are not sure of the reason for this; however, perhaps it is related to the topic of cancer itself, not a likable subject to be tweeted by the lay people.

There is a link for every “replies-to” relationship in a tweet; a tie for each “mentions” connection in a tweet; and a self-loop edge per tweet that is not considered a “replies-to” or “mentions”.

Figure 1 represents a network visualization of the graph’s vertices grouped by cluster using the Clauset–Newman–Moore cluster algorithm [28]. The network graph was laid out using the Harel–Koren fast multiscale layout algorithm. The node size is proportionate to the betweenness centrality score. The edge thickness is related to the reciprocity between users.

To identify the most influential users, top content producers, and disseminators in addition to the main topics and web sources, we conducted SNA using NodeXL [27] and guided by previous research [19,22,23]. To answer RQ1, centrality measure scores were computed: betweenness centrality (BC), in-degree, and out-degree centrality scores [27,28]. To answer RQ2, we identified the top hashtags and word pairs included in tweets. RQ3 was answered by ranking the top shared websites as a source of melanoma-related information.

## 3. Results

Social Network Analysis Results: The network has a low-density score: D = 0.0013. In addition, the average path length score equals 5.3, which means that users are separated by approximately 5 others. The relatively low-density score and average path length imply the slow diffusion of melanoma-related information in this network.

Density between clusters was computed as the total number of ties between two clusters divided by the total possible quantity of ties between them which ranges from 0 = no connectivity to 1 = complete connectivity. Average Path length/Average Geodesic Distance: a path is composed of links connecting two nodes through possibly other nodes in-between [29].

In Figure 1, each small node represents a specific Twitter user. Each time two users interact with each other (e.g., a comment to a tweet), a line is created between them. The clusters/groups in the graph represent the number of users discussing a specific topic and are dependent on how often users mention each other. The size of the nodes is proportionate to their BC scores, which measure this user’s influence on the flow of information to others [30,31].

In Figure 1: Group 1, which is the biggest cluster, discussed that melanoma-related research was impacted by the COVID-19 pandemic; Group 2, which is the second biggest cluster, came out of Australia acknowledging the “World Melanoma Month campaign”; and reporting that there was a decrease in the number of skin cancer-related surgeries and advising people to do a regular “skin check”. Most of the network was composed of isolates, which are individuals who tweeted about melanoma; but did not receive any engagement from others. This might indicate that it is not a likable topic.

To identify the 10 top influencers, we computed BC scores and ranked them from the highest to the lowest. As displayed in Figure 2, melanoma-specific medical and research organization accounts, such as the Melanoma Research Alliance, AIM at the Melanoma Foundation, Melanoma Research Victoria, and a dermatologist represented the top influencers.

In Figure 3, accounts with the highest in-degree centrality scores (sources of information) represent renowned medical and scientific organizations for melanoma, such as the American Academy of Dermatology, the Melanoma Research Alliance, skincancer.org, an academic professor, and a melanoma survivor.

The highest out-degree centrality scores (information disseminators accounts) were for accounts for a mix of medical organizations and individuals from countries other than the United States; these included Italy, Spain, and Australia, as displayed in Figure 4.

In Table 1, as per our results, the top hashtags were mainly discussing the event that May is Melanoma Awareness month. The most frequently employed hashtags during the study period were #Melanoma (*n* = 702), #skincancer (*n* = 117), and #melanomaawarenssmonth (*n* = 89). The top word pairs are displayed in Table 2; which indicates that “skin cancer” (*n* = 65) and “# melanoma, # oncology” (*n* = 43) have highly conversed among Twitter users.

Table 3 provides an overview of the most shared URLs/websites as a source of information. The top highly shared websites were medically specific for melanoma; and included a pharmaceutical company, the American Academy of Dermatology, the Journal of Clinical Oncology, the Mayo Clinic, and the Melanoma Research Alliance.

## 4. Discussion

Principal Findings:

To the best of our knowledge, this research is among the first of its kind to employ SNA to identify the different categories of users playing a critical role in melanoma-related information diffusion on Twitter. During the study period, the top content producers, disseminators, and influencers were mainly from the medical and the research community. A 2022 surveillance study found that influencers in the skin cancer network on Twitter were composed of health professionals, patient advocacy and research organizations [14]. However, our study expanded the analysis to identify the source of content and the disseminators of skin cancer information. We believe that this is crucial for designing awareness campaigns and interventions. Like our study findings, the content of other health contexts on Twitter, such as vaccination, obesity, and COVID-19, was driven predominantly by medical and scientific experts [18,22,32]. However, when information drivers are not from the medical community, the possibility of the spread of misinformation increases; as in case of conspiracy theories pertaining to COVID-19 and wearing masks [20,21].

It is noteworthy to mention that the majority of skin cancer information on Instagram originated from accounts of a non-medical background [32,33]. We highlight the importance of investigating and comparing individuals’ behaviors on different platforms to guide health promotion campaigns in general, and skin cancer awareness campaigns specifically.

Our study findings illustrated that the rate of information dissemination was relatively slow and uneven due to the unconnected users in the network; this aligns with previous research studies on Twitter [14,22]. In previous research, more than 80% of cancer messages in general were not retweeted [34]. Previous research attributes the low diffusion rate of health information, and specifically cancer information to several factors, such as the topic itself, the sender of the message, the format of the content, and finally having emotions involved in the content [35,36,37]. First, in relation to the content source, although there is information from credible accounts with a medical background, previous research proposes that information endorsed by celebrities is disseminated faster on social media [11,38,39]. Health organizations and medical professionals may need to have their scientific content in the form of stories mixed with emotions such as hope; this could positively enhance the information diffusion rate [35].

The top word pairs show that they originated from medical and professional individuals and organizations. The word pairs “#melanoma, #oncology and #new, #article” could highlight new melanoma-related studies that are promoted on Twitter. Another interesting finding is the presence of melanoma-related Tweets in languages other than English; e.g., del, #melanoma; el, #melanoma; these reflect that melanoma is a global public health issue. However, the content lacked focus on preventive behaviors, which comes into line with previous research [12,13].

Another observation is that individuals had a strong interest in the event related to soccer due to the presence of word pairs related to it, such as “día, mundial and mundial, del”. This event could have captured the Twitter users’ attention and we think it might have contributed to the relatively small number of melanoma-related tweets. We recommend that organizations consider timing when designing awareness campaigns to have more reachability and visibility among social media users.

In summary, the main findings identified the sources, disseminators, and the main influencers controlling melanoma-related information; as well as the topics of interest. This information is immensely critical when establishing skin cancer awareness campaigns and interventions.

The differences in users’ interests and melanoma-related conversations across different social media platforms should be used as a guide when developing awareness campaigns, interventions, and customized messages addressing the specific needs of users on each platform.

The study’s strengths and limitations:

The study findings could be helpful for both research and practical implementations. Skin cancer professionals and organizations of medical backgrounds had a potential role as trusted sources, and disseminators of knowledge confined to melanoma. However, there is a need to give more attention and disseminate more information regarding skin cancer preventive behavior on Twitter. We recommend that health organizations and medical professionals collaborate with influencers or celebrities with a high volume of followers for optimum information dissemination during skin cancer awareness campaigns; see, for example, Rahmani et al. [40]. It is also advisable to portray the content in the form of stories complemented with emotions to grab the user’s attention and enhance the dissemination of useful information.

As with other research, the study incurred several limitations. There is no doubt that social media platforms’ content in general and Twitter specifically is constantly evolving and potentially, alter weekly. Another limitation is that the study period is limited to May, which is Melanoma and Skin Cancer Awareness month. Expanding the study period may include seasonal changes that might be useful in future research. Moreover, we only collected tweets in the English language. Finally, as with other studies on social media platforms, we can only collect data from publicly available accounts. Future studies should move to other social media platforms, such as YouTube or TikTok, to uncover characteristics of melanoma-related content and users’ behavior. Future research may include other measurements, especially on the user level, such as reciprocity and tie strength.

## 5. Conclusions

In summary, this study highlights that melanoma-related content, sources, and diffusion patterns are controlled by medical experts and organizations. However, it is crucial to invest this opportunity to provide the public with more content related to preventive behaviors. Finally, we emphasize the importance of studying online behaviors across various social media platforms to gain more insights into users’ needs. This will inform public health officials and maximize the effectiveness of public awareness campaigns; thus, developing customized messages and interventions.

## Figures and Tables

**Figure 1 ijerph-19-11284-f001:**
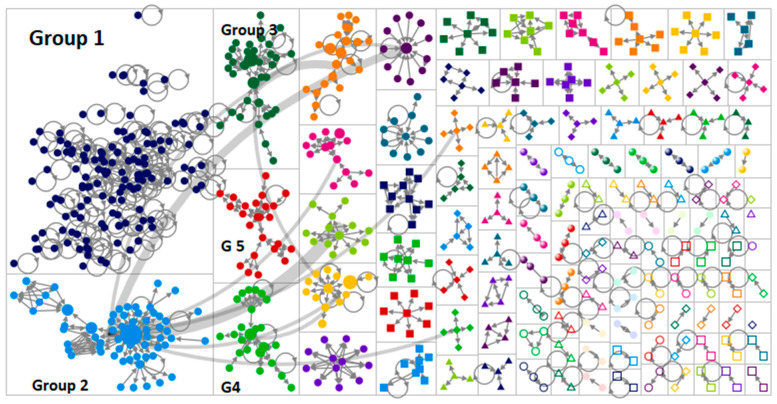
Represents a visualization of melanoma on Twitter.

**Figure 2 ijerph-19-11284-f002:**
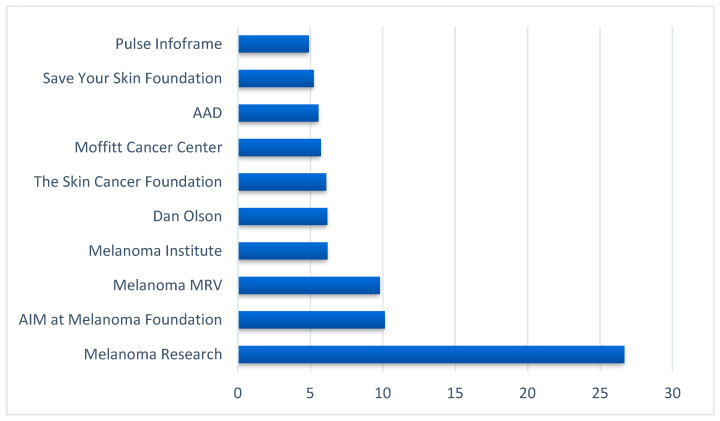
Top Influencers in the melanoma network on Twitter ranked by the betweenness centrality scores.

**Figure 3 ijerph-19-11284-f003:**
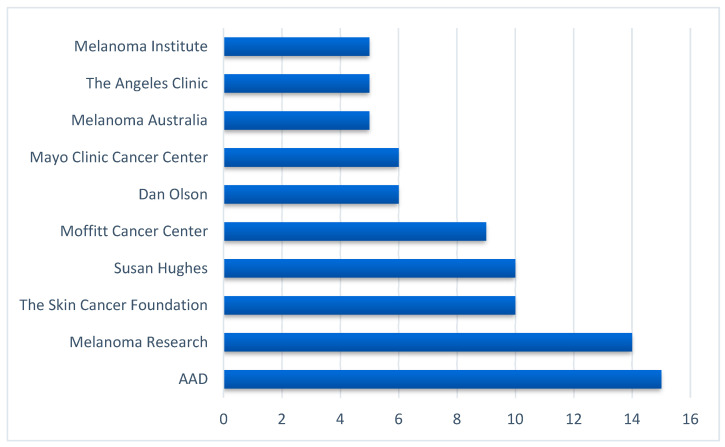
Top sources of melanoma-related information on Twitter ranked by the in-degree centrality scores.

**Figure 4 ijerph-19-11284-f004:**
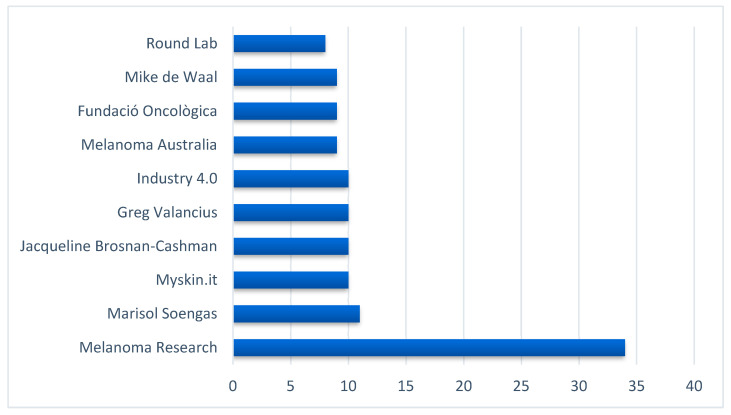
Top disseminators of melanoma-related information on Twitter ranked by the out-degree centrality scores.

**Table 1 ijerph-19-11284-t001:** Top 10 hashtags during the study period.

Top Hashtags	*n*
Melanoma	702
Skincancer	117
melanomaawarenessmonth	89
Oncology	48
skincancerawarenessmonth	45
Melanomamonday	36
Cancer	26

**Table 2 ijerph-19-11284-t002:** Top 10 word pairs during the study period.

Top Word Pairs in Tweet	Count
skin, cancer	65
# melanoma, # oncology	43
new, article	42
del, # melanoma	39
día, mundial	34
# melanoma, # skincancer	27
mundial, del	27
el, # melanoma	24
awareness, month	22
# skincancer, # melanoma	21

**Table 3 ijerph-19-11284-t003:** The top-shared websites shared on Twitter as a source of melanoma-related information.

Top Accounts Tweeted	Type of Account	Number of Re-Tweets
Pfizer Inc.	Pharmaceutical company	11
American Academy of Dermatology	Academic journal	8
Journal of Clinical Oncology	Academic journal	6
American Academy of Dermatology	Academic journal	5
Melanoma Research Alliance	Research organization	4
Prevention.com	Health website	4
Mayo Clinic	Healthcare organization	4
Journal of Clinical Oncology	Academic journal	4
Mobile Foot Clinic	Health website	4

## Data Availability

Data are available upon request.

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
