# Peer review of "Understanding Melanoma Talk on Twitter: The Lessons Learned and Missed Opportunities"

_ijerph, 2022, doi:10.3390/ijerph191811284_

Round 1

Reviewer 1 Report

The topic is interesting and practical. Also, this study offers a good and new insight into understand the current state of melanoma related information exchanged between Twitter users. The research design, questions, hypotheses and methods are clearly stated. And the results are clearly presented.

However, I think some improvements are needed.

(1) Why information diffusion is slow and uneven among users? Maybe it is due to lower density and longer path length. But why? This paper lacks sufficient explanation. Furthermore, Only by understanding the reasons can we propose better measures to promote the diffusion of information.

(2)As the authors pointed, the samples selected will affect the results. Thus, it seems necessary to expand samples.

Reviewer 2 Report

This paper collects Tweets related to Melanoma to discover the top influencers, information sources, disseminators. It also discusses the main topic and users' sharing behavior. The research question is clear and meaningful. However, the research design and writing need great revision.

1.  This paper should add the section of literature review to analyze the status of previous studies. A lot of work has worked on disease by collecting data from online community or social media. All these previous studies could provide research base for this study. Without literature review, it's hard to illustrate the innovation of the paper.

2. Both the data analysis and discussion are simple. It needs to give detailed parameter calculation methods. The figure is not clear. For example, the text of x abscissa in Figure 3 does not match the text of the paper. The data analysis is too descriptive. Because of the lack of multivariate statistical analysis, the findings cannot reveal a deeper principle. Moreover, it should compare the findings with previous studies to show what new things this paper has discovered or how the findings strengthen the previous studies.

Round 2

Reviewer 2 Report

Yes, the revision is improved than original version.  I suggest that the research findings can also be compared with the findings of other diseases based on social media to highlight the uniqueness of skin cancer. I agree that this paper will be accpeted after some minor modifications.

Author Response

We thank the reviewer for this valuable suggestion. We followed the reviewer's recommendation and added the paragraph below to the discussion section.

"Like our study findings, content of other health contexts on Twitter, such as vaccination, obesity and COVID-19 was driven predominantly by medical and scientific experts [18, 22,33] . However, when information drivers are not from the medical community, the possibility of spread of misinformation increase as in case of conspiracy theories pertained to COVID-19 and wearing masks [20, 21]"